# Quality of hypertension management in public primary care clinics in Malaysia: An update

Xin Rou Teh[1]☯*, Ming Tsuey Lim[1]☯, Seng Fah Tong[2], Masliyana Husin[1], Noraziani Khamis[3], Sheamini Sivasampu[1]

1 Centre for Clinical Outcomes Research, Institute for Clinical Research, National Institute of Health (NIH), Ministry of Health, Selangor, Malaysia, 2 Department of Family Medicine, University Kebangsaan Malaysia, Selangor, Malaysia, 3 Center for Clinical Governance Development, Institute for Health Management, National Institutes of Health, Ministry of Health (NIH), Selangor, Malaysia

☯ These authors contributed equally to this work.
* xinrou1801@gmail.com

## Abstract

### Introduction

Adequate control of hypertension is a global challenge and is the key to reduce cardiovascular disease risk factors. This study evaluates management of hypertensive patients in primary care clinics in Malaysia.

### Methods

A cross-sectional analysis of 13 784 medical records from 20 selected public primary care clinics in Malaysia was performed for patients aged ≥30 years old who were diagnosed with hypertension and had at least one visit between 1st November 2016 and 30th June 2019. Multivariable logistic regression adjusted for complex survey design was used to determine the association between process of care and blood pressure (BP) control among the hypertensive patients.

### Results

Approximately 50% of hypertensive patients were obese, 38.4% of age ≥65 years old, 71.2% had at least one comorbidity and approximately one-third were on antihypertensive monotherapy. Approximately two-third of the hypertensive patients with diabetic proteinuria were prescribed with the appropriate choice of antihypertensive agents. Approximately half of the patients received at least 70% of the target indicated care and 42.8% had adequately controlled BP. After adjusting for covariates, patients who received counseling on exercise were positively associated with adequate BP control. Conversely, patients who were prescribed with two or more antihypertensive agents were negatively associated with good BP control.

**Data Availability Statement:** All relevant data are provided within the manuscript. Individual level data cannot be made publicly available due to ethical and patient confidentiality restrictions. The

data collected within this study is managed in accordance with the Personal Data Protection Act in Malaysia and is subject to ethical restrictions under the Medical Research and Ethics Committee, Ministry of Health Malaysia. Data is available on request and interested researchers may address data access requests to the Institute for Clinical Research at contact@crc.gov.my.

**Funding:** This study was supported by a grant from the Ministry of Health, Malaysia (NMRR-17-267-34768) under the Malaysian Health Systems Research initiative. The funders had no role in the study design, data collection and analysis, decision to publish, or preparation of the manuscript.

**Competing interests:** The authors have declared that no competing interests exist.

## Conclusions

These findings indicated that BP control was suboptimal and deficient in the process of care with consequent gaps in guidelines and actual clinical practices. This warrants a re-evaluation of the current strategies and approaches to improve the quality of hypertension management and ultimately to improve outcome.

## Introduction

Hypertension is an important public health challenge as it is associated with higher risk of cardiovascular (CV) and renal diseases [1]. A pooled analysis on worldwide trends in blood pressure (BP) from 1975 to 2015 showed that the number of adults with hypertension had increased from 594 million in 1975 to 1.13 billion in 2015 [2]. As of 2017, hypertension has been identified as the leading risk factor for premature death and disability worldwide [3]. There is an increasing concern that hypertension is becoming more prevalent in low- and middle-income countries than in high income countries [1, 4]. Furthermore, it disproportionately affects populations in low- and middle-income countries with resource constrained health systems [1], thus, achieving optimal BP control is a challenge [5].

The proportion of treated hypertensive patients achieving BP control (<140/90 mm Hg), reviewed ten years ago, were less than 40% in both developed and developing countries [4, 6]. Malaysia like other developing countries is not spared from the alarmingly low rate of BP control among the treated hypertensive patients. The National Health and Morbidity Survey for non-communicable disease risk factors from 2006 to 2015 showed that more than 75% of the hypertensive patients were on antihypertensive treatments [7]. Despite the high proportion of patients received treatment over time, the overall BP control (<140/90 mmHg) was 27.5% in 2006, 34.3% in 2011 and 37.4% in 2015 [7]. In essence, hypertension control in Malaysia still remains inadequate.

One of the possible causes for poor hypertension control could be due to suboptimal quality of hypertension care provided to the patients [8–11]. It was observed that patients who received optimal hypertensive care were more likely to have better BP control [8, 10]. The quality of hypertensive care provided can be assessed by conducting essential care processes received by medical record extraction. This step also includes monitoring of the process of care which are closely related to CV risks such as BP and low-density lipoprotein-cholesterol levels. The selected process measures are usually evaluated against a standard set of criteria as recommended by hypertension management guidelines which include concise evidence-based recommendation to prescribers that have been regularly updated, published and disseminated.

A large majority of hypertensive patients receive their care in primary care setting in Malaysia [12]. A strong coordinating role of primary care is essential to ensure adequate coordination of patient care between primary care providers, as well as coordination of patient care between primary care and other levels of health care to cope with the demand for long term care arrangements. This is of critical importance as evidence has shown that patients with chronic conditions including hypertension are better managed in primary care and this could reduce avoidable hospitalization rates [13].

In Malaysia, improvement in process of care has been undertaken over the years with better establishment of non-communicable disease care such chronic care concepts, diabetic registry, nurse educator and better range of antihypertensive medication. Although there were some evaluations on hypertension, process of care evaluation was last done 10 years ago [11]. Most

of the evaluations on hypertensive care focused on the burden of hypertension in the Malaysian population [14–16]. Similarly, evaluations on process of care and BP control in low- and middle-income countries are scarce. Therefore, we aimed to evaluate the current process of care for hypertension management as well as to explore the association between process of care and BP control among hypertensive patients in public primary care clinics in Malaysia.

## Materials and methods

### Study design, population and sampling

Ethical approval was granted by the Medical Research and Ethics Committee, Ministry of Health Malaysia (NMRR-17-267-34768). Waiver of consent was obtained as medical records were reviewed retrospectively. All records were anonymized before use in the analysis.

Malaysia has a dual tiered healthcare system consisting of government-led and funded public sector as well as private sector with charges fee-for-service. This study focused on the public sector because hypertension was largely managed in this healthcare setting [12]. The evaluation was undertaken based on part of a larger study entitled "Evaluation of the Enhanced Primary Healthcare (EnPHC) interventions in public health clinics" (EnPHC-EVA: Facility). At the point of writing, the study protocol of EnPHC-EVA: Facility is under journal review. The EnPHC-EVA: Facility was a quasi-experimental controlled study which assessed the effectiveness of EnPHC intervention package on process of care and intermediate clinical outcomes for type 2 diabetes mellitus (T2DM) and hypertension across 40 public health clinics located in Malaysia. The criteria to match the 40 selected clinics were i) mean daily attendances ii) the number of medical doctors and family medicine specialists iii) geographical location (urban or rural) and iv) the availability of electronic medical records. As a result, 20 matched pairs were acquired and each clinic within the pairs was randomly assigned to either intervention or control arm using a coin flipping method.

The present analysis utilised the data from patients' medical records in the control clinics reviewed between 1st November 2016 and 30th June 2019. The inclusion criteria were the medical records of all patients aged 30 years and above, diagnosed with hypertension and had at least one visit during the study period. Pregnancy induced hypertension were excluded since their disease management was different. A systematic random sampling was used to sample the patients' medical records in the clinics. Data such as patients' demographic and clinical characteristics were extracted into the electronic structured data collection form using a mobile tablet with validation rules to ensure validity.

### Measures

The measurements of interest were 1) process of care and 2) prescribing practice for hypertensive patients. The process of care included documentation of patients' physical examinations such as systolic BP and diastolic BP, weight, height, waist circumference, body mass index (BMI), electrocardiography test and fundus examinations. Laboratory investigations including blood glucose, serum creatinine, fasting lipid profile, and urinalysis for microalbuminuria were extracted. All physical examinations and laboratory investigations done within the past one year were considered with the exception of weight and BMI measured within past six months as well as BP assessed at the current visit for better accuracy. In addition, process of care on CV risk assessment such as Framingham Risk Score performed in the past one year and types of counseling provided were recorded. Types of counseling extracted included counseling on diet education, exercise/physical activity, smoking cessation, salt intake and alcohol intake. Data on prescription were also extracted which included the types and number of antihypertensive agents. The antihypertensive agents prescribed were categorized into i)

one drug, ii) two drugs and iii) ≥ three drugs. Those without anti-hypertensive agents prescribed but with a diagnosis of hypertension were considered to be on lifestyle modification. The process of care and the prescribing practices that were not documented will be interpreted as "not done" even though the patient may indeed receive the care in actuality [17].

In this study, "controlled hypertension" is defined as having BP <140/90 mmHg (for hypertensive patients without any comorbidity), BP <140/80 mmHg (for diabetic hypertensive patients), BP <130/80 mmHg (for hypertensive patients with ischaemic heart disease /cerebrovascular disease/renal impairment) and BP <150/90 mmHg (for patients aged 80 years and above) as defined by our Malaysian Clinical Practice Guidelines (CPG) on Hypertension 2018 [18]. Patients were categorised as obese and non-obese using cut-off point 27.5 kg/m$^2$ according to the World Health Organisation BMI criteria for Asians [19]. Those antihypertensive drugs that were prescribed not according to criteria set in the Malaysian CPG on Hypertension 2018 were considered as inappropriate choice of antihypertensive agents [18]. These drugs were considered inappropriate choice if any of the following was fulfilled: i) alpha-blocker was prescribed as a single agent to hypertensive patients with T2DM or elderly patients ≥ 65 years old ii) beta-blocker was prescribed as a single agent to hypertensive patients with T2DM or dyslipidemia, iii) diuretic was prescribed as a single agent for hypertensive patients with T2DM or dyslipidemia, iv) angiotensin converting enzyme inhibitor (ACEI) or angiotensin receptor blocker (ARB) was not prescribed to T2DM patients with proteinuria.

## Statistical analysis

Continuous variables were presented as mean and standard deviation while categorical variables were reported in frequencies and percentages. Multivariable logistic regression using complex survey design to account for clustering effect within clinics was used to estimate the adjusted odds ratio and 95% confidence interval for the association of process of care and prescribing practice on BP control. Complete case analysis was performed for the regression. The process of care variables included in the regression were documentation of BMI, electrocardiography, funduscopy, fasting/random blood glucose, creatinine, low-density lipoprotein-cholesterol, urine albumin, Framingham Risk Score, smoking cessation counseling, diet education counseling, salt intake counseling, alcohol intake counseling, and exercise/physical activity counseling. The variable used for prescribing practice was number of antihypertensive agents prescribed. Covariates included for adjustment were age, gender, ethnicity, the number of comorbidities, obesity, the location of primary care clinic and duration of hypertension. Multicollinearity of the covariates was checked. This was a cross sectional study on hypertension management, thus we could not include dropouts and compliance as our covariates. A p-value of <0.05 was considered as statistically significant. Data were analysed using Stata version 14.3 [20].

## Results

Overall, a total of 13 784 patient medical records were reviewed (Table 1). The mean age was 61.0 years, ranging between 30 to 97 years of age. The majority were women (60.4%) and of Malay ethnicity (66.0%) which reflects the local demographic profile. Approximately 50% of hypertensive patients were obese, 38.4% of age ≥65 years old, more than two-third had at least one comorbidity and less than 50% had adequately controlled BP.

Table 2 shows the proportion of documented process of care received by the hypertensive patients. Half of process of care for physical examination and laboratory investigation had at least 70% of the clinical parameters documented. Nevertheless, almost all the records had BP readings on visit date. The laboratory findings indicated a lower proportion of the patients had

**Table 1. Patient demographic data.**

| Characteristic | n(%) |
|---|---:|
| **Mean age, years (SD)** | 61.0(11.1) |
| **Age group** (years) | |
| <65 | 8487(61.6) |
| ≥ 65 | 5297(38.4) |
| **Gender** | |
| Female | 8322(60.4) |
| Male | 5462(39.6) |
| **Ethnicity** | |
| Malay | 9098(66.0) |
| Chinese | 3188(23.1) |
| Indian | 1298(9.4) |
| Others | 200(1.5) |
| **Location of primary care setting** | |
| Urban | 7988(58.0) |
| Rural | 5796(42.0) |
| **Obesity** (n = 10 400) | |
| BMI <27.5 kg/m$^2$ | 5215(50.1) |
| BMI ≥27.5 kg/m$^2$ | 5185(49.9) |
| **Comorbidity** | |
| T2DM | 6901(50.1) |
| Dyslipidaemia | 6643(48.2) |
| Chronic Kidney Disease | 1408(10.2) |
| Ischaemic heart disease | 918(6.7) |
| Stroke/Transient Ischaemic Attack | 583(4.2) |
| Heart failure | 205(1.5) |
| **Number of co-morbidities** | |
| 0 | 3968(28.8) |
| 1 | 4576(33.2) |
| ≥2 | 5240(38.0) |
| **Mean duration of hypertension, years (SD)** | |
| (n = 13 783) | 7.0(6.0) |
| **BP Status (Overall)** | |
| Controlled | 5903(42.8) |
| Uncontrolled | 7881(57.2) |

If n is not stated, the total patients included in the analysis was 13 784.

*SD* standard deviation, *BMI* body mass index, *BP* blood pressure, *T2DM* type 2 diabetes mellitus

documentation of low-density lipoprotein–cholesterol and high-density lipoprotein–cholesterol compared to total cholesterol and triglyceride level. Unfortunately, less than one percent of the records had documented CV risk assessment. Non-pharmacological management including counseling on cessation of smoking, salt intake and alcohol consumption were observed in less than 20% of the records.

The rate of BP control was lower among the hypertensive patients with comorbidities compared to hypertensives alone (Table 3). Hypertensive patients with lifestyle modification appeared to have a better BP control rate compared to those on antihypertensive agents.

**Table 2. Process of care for hypertensive patients.**

| Process of care | n(%) |
|---|---:|
| **Documented Physical examination** | |
| SBP reading in the current visit | 13 674(99.2) |
| DBP reading in the current visit | 13 674(99.2) |
| Weight measurement in the past 6 months | 11 978(86.9) |
| Height measurement | 10 944(79.4) |
| Waist circumference measurement | 8220(59.6) |
| ECG in the past 1 year (n = 13 454) | 5669(42.1) |
| BMI measurement in the past 6 months | 5227(37.9) |
| Funduscopy in the past 1 year (n = 13 545) | |
| With T2DM (n = 6680) | 2274 (34.0) |
| Without T2DM (n = 6865) | 136 (2.0) |
| **Documented laboratory investigation in the past 1 year** | |
| **Blood glucose** | |
| Fasting/random blood glucose | 11 985(86.9) |
| **Renal profile** | |
| Creatinine | 11 448(83.1) |
| **Fasting lipid profile** | |
| Total cholesterol | 11 165(81.0) |
| Triglycerides | 11 072(80.3) |
| HDL cholesterol | 9201(66.8) |
| LDL cholesterol | 9061(65.7) |
| **Urine test** | |
| Urine albumin (n = 13 689) | 8095(59.1) |
| **CV risk assessment in the past 1 year** | 111(0.8) |
| **Counseling** | |
| Diet education | 5998(43.5) |
| Exercise/Physical activity | 4512(32.7) |
| Smoking cessation (n = 10 161) | 1144(11.3) |
| Salt intake | 1093(7.9) |
| Alcohol intake (n = 11 775) | 384(3.3) |

If n is not stated, the total patients included in the analysis was 13 784.

*SBP* systolic blood pressure, *DBP* diastolic blood pressure, *ECG* Electrocardiogram, *BMI* body mass index, *LDL* low-density lipoprotein, *HDL* high-density lipoprotein, *T2DM* type 2 diabetes mellitus, *CV* cardiovascular

Overall, 97.5% of the hypertensive patients were prescribed antihypertensive drugs while the remaining (2.5%) were on lifestyle modification (Table 4). Approximately two-third of the hypertensive patients were on two or more types of antihypertensive agents. When all cases of monotherapy and the combination of two or more antihypertensive agents were considered together, calcium channel blockers (CCBs) (77.2%) were the most commonly prescribed followed by ACEIs (54.8%), beta blockers (29.8%), diuretics (23.4%), ARBs (5.8%) and alpha-blockers (4.6%). Almost all the patients (>98%) with comorbidities were prescribed with the appropriate choice of antihypertensive agents except for hypertensive diabetic patients with proteinuria in which less than 70% were prescribed with ACEI or ARB.

Fig 1 shows the results of multivariable logistic regression analysis which explored the factors associated with BP control. After adjusting for covariates, counseling on exercise was shown to positively associated with BP control (OR 1.31, 95% CI 1.04–1.63). Conversely,

**Table 3. Blood pressure control status in different subgroups of hypertensive patients.**

| BP Status by group as per CPG 2018 | n(%) |
|---|---:|
| **HPT with antihypertensive agents (n = 13446)** | |
| Controlled | 5732(42.6) |
| Uncontrolled | 7714(57.4) |
| **HPT on lifestyle modification (n = 338)** | |
| Controlled | 171(50.6) |
| Uncontrolled | 167(49.4) |
| **HPT without comorbidity and ≤ 80 years (n = 5831)** | |
| Controlled | 3178(54.5) |
| Uncontrolled | 2653(45.5) |
| **HPT with T2DM (n = 4970)** | |
| Controlled | 1751(35.2) |
| Uncontrolled | 3219(64.8) |
| **HPT with CKD/IHD/Stroke/TIA (n = 2466)** | |
| Controlled | 591(24.0) |
| Uncontrolled | 1875(76.0) |
| **Age > 80 years (n = 517)** | |
| Controlled | 383(74.1) |
| Uncontrolled | 134(25.9) |

If n is not stated, the total patients included in the analysis was 13 784.

*SD* standard deviation, *CPG* clinical practice guidelines, *BMI* body mass index, *BP* blood pressure, *TIA* transient ischaemic attack, *IHD* ischaemic heart disease, *CKD* chronic kidney disease, *HPT* hypertension, *T2DM* type 2 diabetes mellitus

hypertensive patients who were prescribed with two or more antihypertensive agents were negatively associated with BP control.

## Discussion

To the best of our knowledge, our study is the first to report on documented process of care for hypertension involving multiple centres in Malaysia using patient medical records. In this study, documentation in patient medical records were used as a proxy to describe the process of care in hypertension management. Overall, approximately 43% of hypertensive patients in this study had adequately controlled BP. The rate of BP control was lower among the hypertensive patients with comorbidities compared to hypertensive alone. The process of care received by the hypertensive patients were suboptimal as more than 50% of them did not receive the process of care that were recommended by the local guideline. A substantial proportion of the hypertensive patients with diabetic proteinuria was not prescribed as per CPG [18]. The results showed that counseling on physical exercise was associated with better BP control rate whereas higher number of medications used was associated with worse BP control rate.

The overall BP control in this study is comparable to that of previous local studies at primary care setting [21, 22]. This showed that there has not been much improvement in the overall BP control rate despite the advancement in process of care arising from the amelioration in care structure in Malaysian public primary care clinics over the years. This includes the availability of trained family physicians, diabetes registry, non-communicable disease units in the clinics, improved laboratory service in the clinic for chronic disease management and availability of a wide range of anti-hypertensives [12, 23–26]. A local study reported that there

**Table 4. Prescribing pattern of antihypertensive agents.**

| Antihypertensive agents prescribed | Total, n(%) | BP status, n(%) | |
|---|---|---|---|
| | | Controlled | Uncontrolled |
| **Number of antihypertensive agents prescribed** | | 5903 | 7881 |
| 0 | 338(2.5) | 171(2.9) | 167(2.1) |
| 1 | 4994(36.2) | 2535(42.9) | 2459(31.2) |
| 2 | 5030(36.5) | 2102(35.6) | 2928 (37.2) |
| $\geq 3$ | 3422(24.8) | 1095(18.5) | 2327 (29.5) |
| **Type antihypertensive agents (n = 13 446)** | | 5732 | 7714 |
| CCB | 10 382(77.2) | 4363(76.1) | 6019(78.0) |
| ACEI | 7363(54.8) | 2754(48.0) | 4609(59.7) |
| Beta-blocker | 4002(29.8) | 1519(26.5) | 2483(32.2) |
| Diuretics | 3152(23.4) | 1166(20.3) | 1986(25.7) |
| ARB | 782(5.8) | 300(5.2) | 482(6.2) |
| Alpha-blocker | 623(4.6) | 196(3.4) | 427(5.5) |
| **Inappropriate choice of antihypertensive agents** | | | |
| **Alpha-blocker as single agent among** | | | |
| T2DM patients (n = 6883) | 5(0.1) | 4(0.2) | 1(0.0) |
| Elderly (age $\geq$65 years) patients (n = 5276) | 6(0.1) | 4(0.2) | 2(0.1) |
| **Beta-blocker as single agent among** | | | |
| T2DM patients (n = 6883) | 101(1.5) | 40(1.8) | 61(1.3) |
| Dyslipidaemia (n = 6624) | 125(1.9) | 63(2.3) | 62(1.6) |
| **Diuretics as single agent among** | | | |
| T2DM patients (n = 6883) | 53(0.8) | 26(1.1) | 27(0.6) |
| Dyslipidemia (n = 6624) | 56(0.8) | 29(1.0) | 27(0.7) |
| **ACEI or ARB not prescribed for T2DM patients with proteinuria (n = 3475)** | 1135(32.7) | 493(40.0) | 642(28.6) |

If n is not stated, the total patients included in the analysis was 13 784.

*ACEI* Angiotensin converting enzyme inhibitor, *ARB* Angiotensin receptor blocker, *CCB* Calcium channel blocker, *T2DM* type 2 diabetes mellitus.

was an increasing trend in the percentage of documented investigations conducted for evaluation of hypertension during a patient's first clinic visit from 1998 to 2012 [17]. Apart from this, the Malaysian Statistics on Medicines report revealed that there was substantial increase in utilisation pattern of certain antihypertensive classes in the public primary care clinics [25]. In terms of prescribing pattern, CCBs were the most prescribed class of antihypertensive followed by ACEIs, beta blockers and diuretics from 2011 to 2014 [25]. Similar antihypertensive prescribing trend was also observed in this study. The high utilisation of CCBs especially amlodipine was due to the removal of prescribing restriction in the Ministry of Health Malaysia formulary as well as the introduction of generic amlodipine in the public primary care clinics [27]. In addition, beta-blockers which were once recommended as the first-line treatment for hypertension was later superseded by other classes of antihypertensive drugs [18]. Thus, the possible factors contributing to suboptimal control of hypertension could be due to patient-related factors including patients' lifestyle and other risk factors, difficulties in adherence to prescribed regimens, limited access to care and lack of knowledge about seriousness of hypertension. It can also be influenced by healthcare provider-related factor such as effectiveness of counseling on lifestyle factors and self-care [28–30]. The suboptimal coordination of hypertension care that were received by our patients may be another possible cause for the poor BP control. Although the Chronic Care Model [31] has provided the structure needed for managing

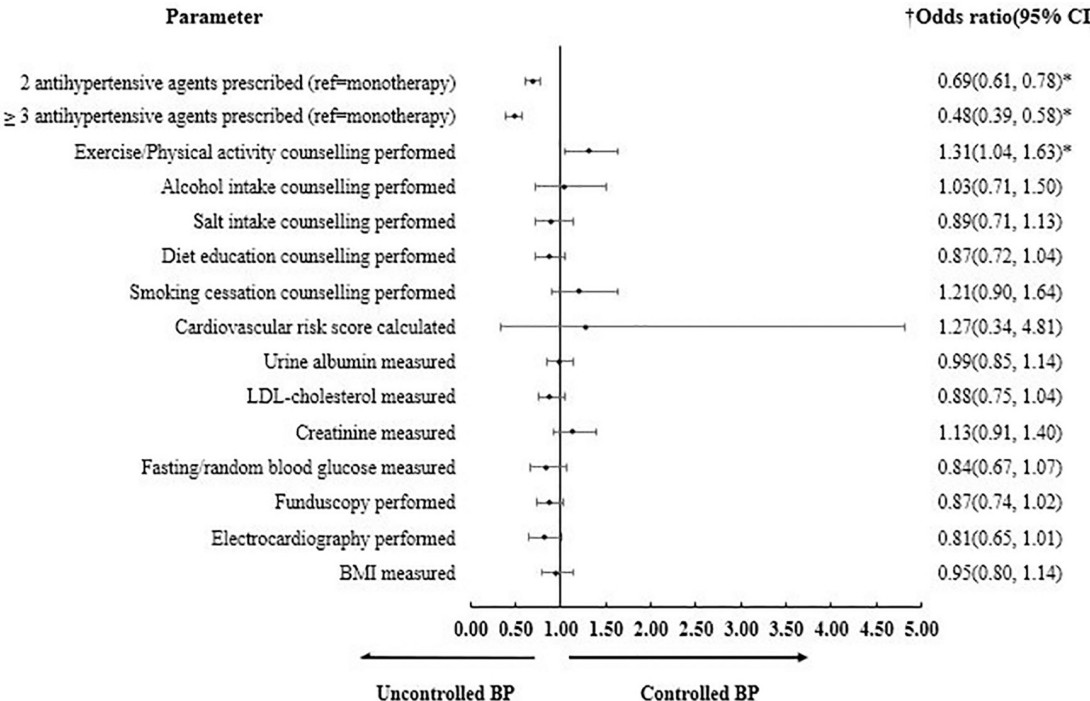

**Fig 1. Adjusted ORs with corresponding 95% CI for factors associated with BP control.** *OR* odds ratio, *CI* confidence interval, *BP* blood pressure. † Adjusted for age, gender, ethnicity, number of comorbidities, obesity, location of primary care clinic (urban or rural) and duration of hypertension. *p<0.05.

chronic disease such as hypertension, the effective implementation still requires optimal coordination, consultation, counseling and good communication skills. However, these need to be evaluated in future studies.

With regards to suboptimal process of care based on the CPG [18], the fact that almost all hypertensive patients did not receive complete CV risk assessment is of great concern. The incomplete or inadequate assessment of CV risk factors may result in missed opportunities for early intervention and pose a negative impact on the morbidity and mortality of the patients. A decrease in these CV risk factors has been shown to reduce CV morbidity and mortality in both individuals with or without established CV disease [32]. Furthermore, the data from the National Health and Morbidity Surveys of Malaysia in 2015 [7] indicated that the prevalence of CV risk factors including hypertension, dyslipidaemia, T2DM, overweight or obesity and smoking is high and has been on an increasing trend. The possible causes for the poor rate of compliance to guidelines or the practice gap for the CV risk assessment were reported to be the absolute lack of physician time [33], knowledge-related factors such as lack of awareness, clinical experience and familiarity or attitude-related factors including lack of agreement, lack of outcome expectancy, self-efficacy, and motivation [29]. Nevertheless, important CV risk such as documentation of investigation on fasting blood glucose, cholesterol profile, renal function, BP status in this study were well documented.

Another important finding worth noting is that adequacy rate of fundoscopy was similarly observed in a previous local clinical audit on hypertensive care [9]. This indicated that a high proportion of the hypertensive patients with T2DM have not undergone regular eye examinations in the primary care clinics as recommended by clinical guidelines [34]. Regular eye examination is essential since early detection and treatment of eye diseases including

retinopathy considerably reduce the incidence of blindness. This is because early retinopathy tends to be unnoticeable by patients due to the absence of visual loss in the early stage. In fact, retinopathy is the most common microvascular complication of diabetes mellitus [35]. A study by Erden et al. (2012), indicated that the increase in the incidence of retinopathy is related to the degree of severity and duration of hypertension [36]. The low percentage of patients who had undergone fundus examinations seen may contributed by the lack of awareness among healthcare providers on the needs for eye screening, non-adherence to the guideline, patients defaulting on follow-up examinations, overcrowding at public health clinics or healthcare provider not proficient on the use of direct ophthalmoscopes to examine the fundus or patients' refusal for pupil dilatation [9].

Our findings suggest that there is a large gap between guidelines and actual clinical practices when pharmacological treatment is concerned. Approximately 30% of the diabetic patients with proteinuria were not prescribed with the recommended ACEI or ARB where these antihypertensive classes are the first line treatment options in the absence of any other contraindications [18]. Though there could be valid reasons including cost, side-effects and patient preference for not prescribing ACEI or ARB, the 30% of non-adherence to CPG [18] may be too high to be accountable for by these factors. The medication adherence to guideline issue was also earlier reported in other local studies conducted in primary care settings [22, 28, 30]. For example, according to Cabana et al. (1999), doctors' intentions to use the guidelines can be predicted from their attitudes towards the guidelines, which are influenced by many factors including their own knowledge, past clinical experience, beliefs and adherence to guidelines, outcome expectations, peers' opinions as well as external barriers including patient's characteristics and environmental factors [29]. Therefore, remedial measures should be taken to allow better promotion and adoption of practice guidelines into daily practice of the doctors in the primary care clinics.

Our findings indicate that hypertensive patients who had documented counseling on exercise/physical activity had 1.3 odds of having their BP controlled compared to those not receiving counseling. However, the information on how counseling was performed was not available in the medical record. We assume that the counseling was done as the usual practice in the clinic. The association of exercise/physical activity with lower BP was also reported in a previous study [37]. Aside from giving more exercise prescriptions, doctors may encourage the patients to be more active including walking, using bicycles, climbing stairs or pursuing other means of integrating physical activity into their daily routines. Furthermore, exercise prescription can also be used to manage the obese hypertensive patients. A previous study [38] reported that the obese tend to benefit from a regular exercise regimen in terms of having an improved insulin sensitivity, better lipid and lipoprotein profile, BP as well as a reduced risk of death. Exercise therapy should be performed in lieu with diet therapy to improve obesity. Surprisingly, in this study, no association between counseling on salt intake and good BP control was observed. These findings suggest that counseling on salt intake, although documented, may not be received by patients effectively. Nevertheless, this type of counseling is essential as a study [39] has shown that reduction of salt intake can reduce BP and decrease the need for medications in patients who are "salt sensitive" where BP fluctuations are highly dependent on the level of sodium intake.

In this study, approximately two-third of the hypertensive patients who are on combination therapy still have poor BP control. Conversely, we found that hypertensive patients who have been prescribed with a combination therapy of antihypertensive agents are more likely to have poor BP control. Similar finding was also observed in a previous local study conducted on hypertensive control among hypertensive patients with T2DM in primary care clinics [21]. This unexpected finding is in contrast with the literature where it has been reported that

combination of antihypertensive drugs from different classes have additive or synergistic effects when used together, whereby the reduction in BP is greater than if patients were treated with either drug class alone [40]. It is plausible that patients who are on combination therapy were the subgroup of those who had more resistant BPs or may even have underlying secondary hypertension or there was a failure of physicians to increase the dose, number of antihypertensive agents or change the treatments [21, 41]. Additionally, treating to target with clear agreed target BP may pose as a challenge since disagreement of dosing and number of medications between patients and doctors often occur. This will also contribute to non-adherence and suboptimal BP control.

The major strength of the study is the large sample of patients which allowed sufficient statistical power to explore how the care processes of hypertension affect BP control among hypertensive patients. Another strength was that the study was conducted across 20 chosen primary care clinics which are representative of the public clinics in Malaysia. There were some limitations in this study. First, the cross-sectional study design cannot be used to establish causal and effect relationship between the factors and outcome. Second, since it is a record-based study, we did not have some of the patients' information including the level of education, occupation or socioeconomic status which may influence patients' self-management such as medication adherence or compliance to lifestyle intervention to a certain extent. Third, the information on counseling was limited to type of counseling documented in the medical records. It would be difficult to capture the effectiveness of counseling given to the patients during the doctor-patient consultation session. Thus, we assumed that counseling was delivered accordingly by the attending primary care doctors. Similarly, information on the dose and duration of the antihypertensive agents were not extracted because it was assumed that the dose titration process will occur over time, aiming at BP control. Hence, it is not feasible to measure antihypertensive medication titration on BP control with our cross-sectional study design. Fourth, the assumption of the process of care and prescribing practices not documented in the medical records are considered as not done and may underestimate the actual quality of care received by the patients. Lastly, the findings cannot be generalizable to the private clinics since the study population was solely from public health clinics.

## Conclusion

Our present findings show that BP control was suboptimal in public primary care clinics. The process of care remains deficient with gaps in guidelines and actual clinical practice. These suboptimal BP control and hypertensive management findings were consistent with several local cross-sectional surveys and clinical audits on hypertension management conducted in primary care [7, 9, 11, 17, 21, 22, 28, 30]. These consistent findings as previous studies demonstrated that the hypertensive care in Malaysia still remains much as it was in the past decade. Therefore, a continuous and concerted efforts are warranted to improve the quality of hypertensive management. Apart from looking at traditional issues of physician awareness of hypertension and physician adherence to guidelines, an evaluation on the how the process being undertaken is important in order for us to understand the dynamics within a primary care clinic.

## Acknowledgments

The authors would like to thank the Director General of Health Malaysia for the permission to publish this paper. We would also like to thank the patients and health providers who participated in the EnPHC-EVA: Facility study as well as the EnPHC-EVA: Facility study investigators.

## Author Contributions

**Conceptualization:** Xin Rou Teh, Ming Tsuey Lim, Seng Fah Tong, Sheamini Sivasampu.

**Data curation:** Xin Rou Teh, Ming Tsuey Lim, Masliyana Husin, Noraziani Khamis.

**Formal analysis:** Xin Rou Teh, Ming Tsuey Lim, Seng Fah Tong.

**Funding acquisition:** Sheamini Sivasampu.

**Methodology:** Xin Rou Teh, Ming Tsuey Lim, Seng Fah Tong, Sheamini Sivasampu.

**Project administration:** Sheamini Sivasampu.

**Supervision:** Sheamini Sivasampu.

**Visualization:** Xin Rou Teh, Ming Tsuey Lim.

**Writing – original draft:** Xin Rou Teh, Ming Tsuey Lim.

**Writing – review & editing:** Xin Rou Teh, Ming Tsuey Lim, Seng Fah Tong, Masliyana Husin, Noraziani Khamis, Sheamini Sivasampu.

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
