## [Decision Letter · Decision Letter 0]

1 Apr 2020

PONE-D-19-33878

Quality of hypertension management in public primary care clinics in Malaysia: an update

PLOS ONE

Dear Miss Teh,

Thank you for submitting your manuscript to PLOS ONE. After careful consideration, we feel that it has merit but does not fully meet PLOS ONE’s publication criteria as it currently stands. Therefore, we invite you to submit a revised version of the manuscript that addresses the points raised during the review process.

We would appreciate receiving your revised manuscript by May 16 2020 11:59PM. To enhance the reproducibility of your results, we recommend that if applicable you deposit your laboratory protocols in protocols.io, where a protocol can be assigned its own identifier (DOI) such that it can be cited independently in the future. For instructions see: http://journals.plos.org/plosone/s/submission-guidelines#loc-laboratory-protocols

We look forward to receiving your revised manuscript.

Kind regards,

Anderson Saranz Zago, PhD

Academic Editor

PLOS ONE

Additional Editor Comments (if provided):

According to the opinion of the reviewers, the manuscript brings an interesting subject, however, it needs to be reviewed on several topics. After all these changes, the authors can resubmit the manuscript for a new evaluation.

Sincerely

Anderson Saranz Zago, PhD.

Academic editor

Journal Requirements:

2. Please state the date upon which the data used in this survey was accessed. In addition, please refer to any post-hoc corrections to correct for multiple comparisons during your statistical analyses. If these were not performed please justify the reasons. Please refer to our statistical reporting guidelines for assistance (https://journals.plos.org/plosone/s/submission-guidelines.#loc-statistical-reporting).

Reviewers' comments:

Reviewer's Responses to Questions

**Comments to the Author**

1. Is the manuscript technically sound, and do the data support the conclusions?

Reviewer #1: Partly

Reviewer #2: Yes

2. Has the statistical analysis been performed appropriately and rigorously? 

Reviewer #1: No

Reviewer #2: Yes

3. Have the authors made all data underlying the findings in their manuscript fully available?

Reviewer #1: Yes

Reviewer #2: Yes

4. Is the manuscript presented in an intelligible fashion and written in standard English?

Reviewer #1: Yes

Reviewer #2: Yes

5. Review Comments to the Author

Reviewer #1: This article investigated the management of hypertensive patients in primary care clinics in Malaysia, based upon data from 13,784 medical records. Although the study is sound and data are valuable, some issues should be reviewed.

Introduction

1) The authors provide general information about BP management worldwide and in developing countries, but the central problem remained unclear. What is the central point to be demonstrated and why? In short, the problem and purpose of the work should be clearly defined.

Methods

2) It seems that the only measures of BP management were "types of counselling provided as well as the prescribed antihypertensive medications". How the types of counselling have been effectively assessed and what about the quantity and quality of these interventions? As for the medications, what information has been extracted (type, dose etc). Please detail these aspects.

3) What independent comparisons have been performed through the t-tests and Mann-Whitney tests? What were the processes of care and prescribing patterns inserted in the multivariate analysis?

4) "Complete case analysis was performed for the regression" - did the authors considered the use of intention to treat analysis? Information about the drop outs within the more than 13,000 cases would be useful. How many patients did not comply to the prescribed treatament or failed to perform reasssessment during the study?

Results

1) Overall, although interesting, data seem to reproduce information extensively acknowledged in the literature. This is an important issue to be commented by the authors. Moreover, some aspects must be further clarified. For instance, how "Inappropriate choice of antihypertensive agents" was defined?

2) Only 2.5% of patients received counselling in regards to lifestyle modifications, which seems to be a very small proportion to allow statements about the effectiveness of these procedures to improve BP control. Moreover, information about the characteristics of patients that received this type of counselling would be useful to a better understanding of these findings.

3) Finally, please explain why to take two or more antihypertensive agents was inversely associated with BP management - is this due to the treatment or to the fact that those patients had higher BP levels. The way the text is written may lead to misinterpretation of these findings.

Discussion

1) The statement that the study reported "on the quality of indicated care processes received by hypertensive patients" poses problem. The quality of counselling and other procedures have been not assessed, but only their descriptive characteristics.

2) "The possible factors contributing to suboptimal control of hypertension could be due to patient-related factors including patients’ lifestyle and other risk factors, difficulties in adherence to prescribed regimens, limited access to care and lack of knowledge about seriousness of hypertension". This is true and significangtly limits the interpretation of the presented findings. Additional information on these features should be provided. The use of intention to treat approach could also enhance the strength of the data and should be considered by the authors, or at least information on drop outs should be provided.

3) Although interesting as a text, most part of the Discussion seems to be highly speculative (please refer to lines 242 to 260).

4) The authors seemed to assume the physical activity counselling has been effectively resulted in actual practice - any information on this?

Reviewer #2: The authors managed to demonstrate how the number of hypertensive patients worldwide is growing, and the control of blood pressure in countries still leaves a desire. Therefore, it has a high relevance in the study of evaluating the quality of care in primary care clinics in Malaysia. The article shows solid results, however some corrections are needed.

6. PLOS authors have the option to publish the peer review history of their article (what does this mean?). If published, this will include your full peer review and any attached files.

Reviewer #1: No

Reviewer #2: No

---

## [Author Response · Author response to Decision Letter 0]

25 Apr 2020

Please kindly refer to the document named "Response to Reviewers" for our response to reviewers and editor comments.

---

## [Decision Letter · Decision Letter 1]

11 Jun 2020

PONE-D-19-33878R1

Quality of hypertension management in public primary care clinics in Malaysia: an update

PLOS ONE

Dear Dr. Xin Rou Teh

Thank you for submitting your manuscript to PLOS ONE. After careful consideration, we feel that it has merit but does not fully meet PLOS ONE’s publication criteria as it currently stands. Therefore, we invite you to submit a revised version of the manuscript that addresses the points raised during the review process.

As you can see, just one reviewer asked for minor revision.

We look forward to receiving your revised manuscript.

Kind regards,

Anderson Saranz Zago, PhD

Academic Editor

PLOS ONE

Additional Editor Comments (if provided):

According to the opinion of the one reviewers, the manuscript still have a few points to be adressed. After all these changes, the authors can resubmit the manuscript for a new evaluation.

Sincerely

Anderson Saranz Zago, PhD.

Academic editor

Reviewers' comments:

Reviewer's Responses to Questions

**Comments to the Author**

1. If the authors have adequately addressed your comments raised in a previous round of review and you feel that this manuscript is now acceptable for publication, you may indicate that here to bypass the “Comments to the Author” section, enter your conflict of interest statement in the “Confidential to Editor” section, and submit your "Accept" recommendation.

Reviewer #1: All comments have been addressed

Reviewer #2: All comments have been addressed

2. Is the manuscript technically sound, and do the data support the conclusions?

Reviewer #1: Yes

Reviewer #2: Yes

3. Has the statistical analysis been performed appropriately and rigorously? 

Reviewer #1: Yes

Reviewer #2: Yes

4. Have the authors made all data underlying the findings in their manuscript fully available?

Reviewer #1: No

Reviewer #2: Yes

5. Is the manuscript presented in an intelligible fashion and written in standard English?

Reviewer #1: Yes

Reviewer #2: Yes

6. Review Comments to the Author

Reviewer #1: The manuscript has substantially improved after review and I commend the authors for their effort to address my comments. However, I would make a last suggestion, given the changes made in the Introduction section.

Please rephrase the new hypothesis in lines 85-86: “We hypothesized that hypertensive patients who received care for hypertension were associated with better BP control”. This seems quite obvious as an hypothetical approach and does not reflect the main purpose or results of your study.

In my opinion, the main contribution of this cross-section study is the description of the characteristics of care processes in a substantial amount of public health clinics. Moreover, the odds of specific intervention strategies to produce effective results in terms of BP management were analyzed through logistic regression – this is an important aspect, which has nothing to do with the presented hypothesis.

As you state in lines 165-166, “this was a cross sectional study on hypertension management” and its strong points should be put in value when writing the objectives and related hypotheses.

Reviewer #2: The authors accepted and agreed with the suggestions previously made correctly, provided all the data that support the conclusion of the present study. The statistical analyzes met the proposal of the study, and presented interesting results.

7. PLOS authors have the option to publish the peer review history of their article (what does this mean?). If published, this will include your full peer review and any attached files.

Reviewer #1: Yes: Paulo Farinatti

Reviewer #2: No

---

## [Author Response · Author response to Decision Letter 1]

11 Jun 2020

Dear Editor and Reviewer #1,

The current study aimed to assess the process of care for hypertension management and the association between process of care and blood pressure control in primary care clinics in Malaysia.

INTRODUCTION

The authors were asked to rephrase the hypothesis in lines 85-86: “We hypothesized that hypertensive patients who received care for hypertension were associated with better BP control”. The reviewer felt that it did not reflect the main purpose or results of the study. The authors agree with the reviewer and have omitted the hypothetical statement. It now reads:

Introduction, Line 78-86, page 5:

“In Malaysia, improvement in process of care has been undertaken over the years with better establishment of non-communicable disease care such chronic care concepts, diabetic registry, nurse educator and better range of antihypertensive medication. Although there were some evaluations on hypertension, process of care evaluation was last done 10 years ago [11]. Most of the evaluations on hypertensive care focused on the burden of hypertension in the Malaysian population [14–16]. Similarly, evaluations on process of care and BP control in low- and middle-income countries are scarce. Therefore, we aimed to evaluate the current process of care for hypertension management as well as to explore the association between process of care and BP control among hypertensive patients in public primary care clinics in Malaysia.”

---

## [Decision Letter · Decision Letter 2]

21 Jul 2020

Quality of hypertension management in public primary care clinics in Malaysia: an update

PONE-D-19-33878R2

Dear Dr. Xin Rou Teh

We’re pleased to inform you that your manuscript has been judged scientifically suitable for publication and will be formally accepted for publication once it meets all outstanding technical requirements.

Kind regards,

Anderson Saranz Zago, PhD

Academic Editor

PLOS ONE

Additional Editor Comments (optional):

After resubmission, it can be observed that all comments were addressed. So, I am pleased to inform you that your manuscript has been deemed suitable for publication in PLOS ONE. Congratulations!

Sincerely

Reviewers' comments:

Reviewer's Responses to Questions

**Comments to the Author**

1. If the authors have adequately addressed your comments raised in a previous round of review and you feel that this manuscript is now acceptable for publication, you may indicate that here to bypass the “Comments to the Author” section, enter your conflict of interest statement in the “Confidential to Editor” section, and submit your "Accept" recommendation.

Reviewer #1: All comments have been addressed

2. Is the manuscript technically sound, and do the data support the conclusions?

Reviewer #1: Yes

3. Has the statistical analysis been performed appropriately and rigorously? 

Reviewer #1: Yes

4. Have the authors made all data underlying the findings in their manuscript fully available?

Reviewer #1: Yes

5. Is the manuscript presented in an intelligible fashion and written in standard English?

Reviewer #1: Yes

6. Review Comments to the Author

Reviewer #1: (No Response)

7. PLOS authors have the option to publish the peer review history of their article (what does this mean?). If published, this will include your full peer review and any attached files.

Reviewer #1: **Yes: **Paulo Farinatti